# Intracellular Organization of Proteins and Nucleic Acids via Biomolecular Condensates in Human Health and Diseases

Raffaella Gallo 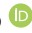

Department of Experimental and Clinical Medicine, University "Magna Græcia", Viale Europa, 88100 Catanzaro, Italy; r.gallo@unicz.it

**Abstract:** Eukaryotic cells are intracellularly divided into several compartments that provide spatiotemporal control over biochemical reactions. Phase separation of proteins and RNA is emerging as an important mechanism underlying the formation of intracellular compartments that are not delimited by membranes. These structures are also known as biomolecular condensates and have been shown to serve a myriad of cellular functions, such as organization of cytoplasm and nucleoplasm, stress response, signal transduction, gene regulation, and immune response. Here, the author will summarize our current understanding of intracellular phase separation, its biological functions, and how this phenomenon is regulated in eukaryotic cells. Additionally, the author will review recent evidence of the role of biomolecular condensates in the development of pathophysiological conditions, with special emphasis on cancer and immune signaling.

**Keywords:** biomolecular condensates; PTMs; cancer and immune response

## 1. Introduction

Eukaryotic cells are extremely busy places. In order to carry out a precise biological function, macromolecules need to partition with their interaction partners in specific compartments. A powerful tool for the intracellular compartmentalization of both proteins and nucleic acids is offered by so-called biomolecular condensates or, simply, condensates. These structures form via a well-studied phenomenon in polymer chemistry called phase separation [1–3] Although the process that underlies the biogenesis of condensates in cells has frequently been referred to as liquid–liquid phase separation (LLPS), scholars have recently argued against the use of this term, as the coexisting phases are not simple liquids but complex viscoelastic network fluids [4]. Therefore, throughout this review, the more general term "phase separation" will be used, which keeps in consideration the complex physical properties of both condensates and surrounding milieu in living cells.

Briefly, above a saturation concentration, homogeneously-distributed polymers in a mixture with a solvent can demix when homotypic interactions are sufficiently energetically favorable to overcome the entropy-driven mixing. As a consequence, a polymer-enriched or condensed phase and a polymer-depleted or dilute phase will form [3]. For biological polymers, such as proteins or nucleic acids in cells, phase separation is achieved via well-regulated changes in protein/RNA concentration, changes in charge due to post-translational modifications, and osmotic or temperature alterations [5]. In cells, the resulting condensates exhibit viscosity and viscoelasticity often compatible with a liquid-like state, adopt round morphologies, display rapid liquid-like merging, and exhibit dynamic exchange with the surrounding nucleoplasm or cytoplasm [5–8].

Intracellular biomolecular condensates are highly multicomponent systems with remarkably diverse composition, and hundreds, if not thousands, of macromolecules constitute them [9–14]. Of those macromolecules, only a small subset, known as "scaffold", is necessary to promote condensate formation and maintain its integrity. Other condensate components, identified as "clients", are recruited to the same compartment but are dispensable for condensate assembly [15]. Scaffolds are typically molecules that have a high

valency, and therefore engage in a large number of interactions, and often present an unstructured regions. The scaffold and client model has been initially investigated by building a simple in vitro model system composed of multivalent polySUMO-polySIM proteins and low valency GFP-SUMO or GFP-SIM proteins [15]. In this system, the high-valency proteins act as a scaffold that recruit the low valency clients depending on the stoichiometries of the scaffold components. In addition, it has been shown that phosphorylated Nephrin is able to recruit Nck and N-WASP, in a manner that depends on the high valency of Nephrin. In this case, phosphorylated Nephrin acts as a scaffold and is highly negatively charged, therefore recruiting the positively charged clients [16]. Centrosomes, which play an important role in the formation of mitotic spindles during mitosis, are biomolecular condensates, formation of which is driven by the scaffold protein SPD-5. Microtubule-associated proteins (MAPs) and tubulins are recruited to the condensates via interactions with SPD-5 and act as clients [17]. Additional examples of scaffold proteins of specific biomolecular condensates are offered by FUS, HNRNPA1, HNRNPA2, ESWR1, and TAF15 [18–20]. Although clients do not drive phase separation, they can play an important role in modulating condensate functions in cells in response to different stimuli, since client composition seems to vary as a consequence of modifications of the multivalent scaffolds [15]. Considerable effort has been made to understand the common features of molecules that form biomolecular condensates. It has been shown that different types of multivalent interactions contribute to phase separation, i.e., protein–protein, protein–RNA, and RNA–RNA interactions between ordered domains and weak, transient, multivalent interactions usually between intrinsically disordered regions (IDRs). The last type of interactions includes cation–anion interactions, π–π interactions, dipole–dipole interactions, and cation–π interactions [21] (Figure 1).

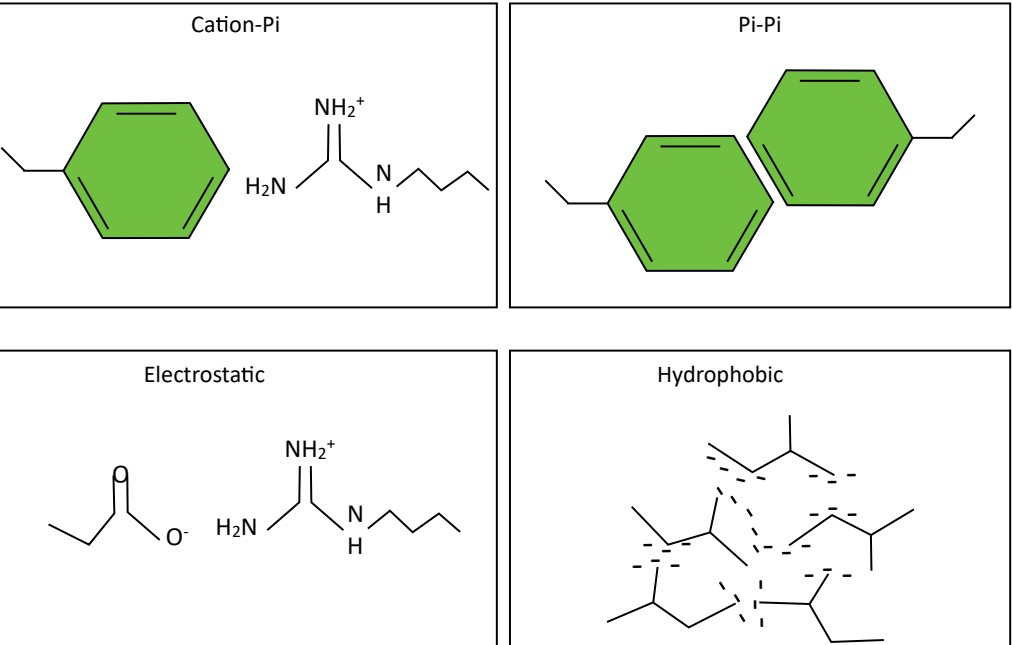

**Figure 1.** Biomolecular condensate−promoting features. The most common types of interaction among biomolecules that contribute to the formation of biomolecular condensate are illustrated. They include cation–π, π−π, electrostatic, and hydrophobic interactions.

In this review, the author will first summarize the different biological roles that have been attributed to intracellular biomolecular condensates. Moreover, an important body of work has focused on the modulation of assembly/disassembly of condensates. The author will therefore describe crucial findings on the tools and molecular regulators used by living cells to regulate phase separation. Finally, it is becoming evident that abnormal phase separation can perturb cellular activities with important consequences for diseases

onset. Therefore, the author will summarize the involvement of aberrant condensates in pathological states such as dysregulated immune responses and cancer.

## 2. Physiological Functions of Biomolecular Condensates

Biomolecular condensates are highly diverse in their physical properties, molecular composition, and subcellular location (Figure 2A). This diversity is also reflected in the number of functions that cellular condensates play (Figure 2B).

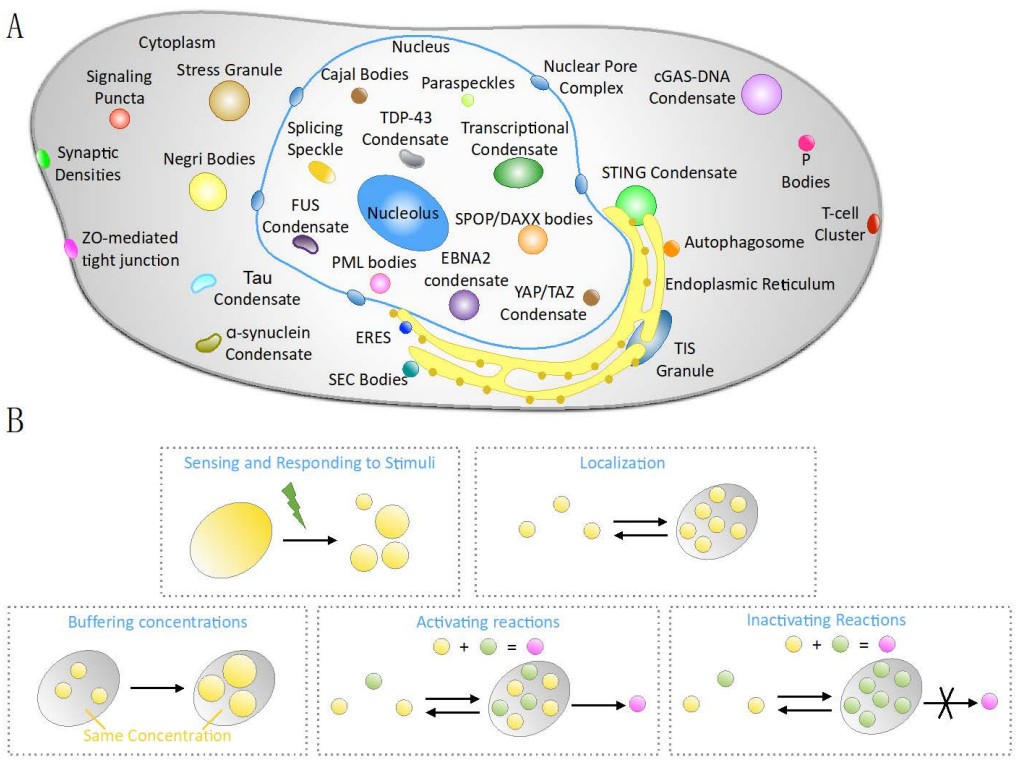

**Figure 2.** Biomolecular condensates and their functions. (**A**) Schematic diagram depicting the most common biomolecular condensates found in eukaryotic cells, including those that are altered or formed in pathological conditions. (**B**) Main functions played by biomolecular condensates: sensing and responding to stress stimuli, promoting the correct localization of biomolecules, keeping biomolecule concentration constant, promoting chemical reactions, and inhibiting reactions by sequestering biomolecules.

Phase separation provides a powerful tool to overcome the large space for protein diffusion in cells and concentrating components into condensates to facilitate biochemical reactions. The concentration of certain macromolecules condensates has been shown to be up to two orders of magnitude higher in condensates than in the surrounding dilute phase [22]. As a consequence, specific enzymatic reactions can occur at higher rates within the condensates. This has been shown for the cleavage of RNA in vitro [23–25], for the melting of double-stranded nucleic acids [24,26], and for transcription of small nuclear (sn)RNA in vivo [27].

Cellular condensates can be exploited by the cell to buffer concentration of proteins and RNA [28]. When the total concentration of macromolecules changes, condensates will change in size and/or number, but the concentration of the macromolecules outside of the condensates will be constant. This has been experimentally demonstrated using an engineered version of DDX4 containing a duplicated ammino-terminal domain [29] and is thought to be true for many nuclear and cytoplasmic biomolecular condensates, such as splicing speckles, promyelocytic leukemia protein (PML) bodies, and processing bodies (P bodies) [30–33].

Because phase separation is a switch-like event and condensates can rapidly form and disassemble, another important function attributed to biomolecular condensates is represented by the ability to sense and respond to stimuli [34,35]. An example is offered by the formation of stress assemblies, such as Stress Granules (SGs) that reversibly concentrate and store translationally-stalled mRNAs, associated pre-initiation factors, and RNA-binding proteins until the stress is resolved [36]. Several studies support the idea that these condensates assemble and disassemble through tightly regulated phase separation [7,34,35,37]. Similarly, in Drosophila S2 cells, amino acid starvation leads to the condensation of trafficking proteins in biomolecular condensates, known as Sec bodies [38–41].

Phase separation is also implicated in transcriptional regulation, both at the level of activation and repression [42–46]. Transcription condensates in the nucleus are often found at the promoter region and at super-enhancers, comprise of transcription factors and coactivators [47–50], and can regulate the transcription of key genes such as Oct4, Sox2, Nanog, Klf4, n-Myc, Utf1, Tet1, Tet2 and other genes that play a prominent role in embryonic stem cell biology and determine cell identity [43,51].

Phase separation is involved in immune responses at different levels. Phase separation mediates T-cell receptor (TCR) signal transduction, a process that is fundamental for T-cell activation and adaptive immune response [52]. After being phosphorylated, the cytoplasmic domains of TCR will recruit specific molecules involved in T-cell activation through several downstream signaling pathways [53]. Investigation of the reconstituted clusters revealed that these structures arise from phase separation, in order to generate compartments that are enriched in kinases but excluded phosphatases, thereby facilitating signaling [54].

Another example of the involvement of biomolecular condensates in immune signaling is offered by the cGAS-STING pathway. The DNA sensor Cyclic GMP–AMP synthase (cGAS) is a pattern recognition receptor (PRR) that binds the DNA of pathogens and catalyzes the production of cyclic GMP-AMP (cGAMP), leading to the activation of the adaptor protein of interferon genes (STING) [55]. Recently, it has been shown that cytosolic DNA, such as viral DNA and cGAS condensates, are formed by the multivalent interactions between a disordered and positively charged region cGAS and DNA via phase separation [56]. The resulting condensates are thought to concentrate the ATPs and GTPs together with activated cGAS to efficiently produce cGAMP. This event initiates anti-viral immune responses via cGAMP-mediated activation of STING and downstream signaling that promotes the expression of type I interferons and other proinflammatory cytokines [57,58]. In this system, presence of condensates has an anti-viral effect.

## 3. Regulation of Phase Separation

How condensate assembly/disassembly, composition and biophysical properties are modulated in complex biological systems is an outstanding question, which is only beginning to be addressed.

### 3.1. Biophysical Parameters

Alterations of biophysical parameters impact condensates formation. Concentration of condensates components is an important factor that can be modulated by a variety of ways, such as biosynthesis, degradation and transport of those components [59]. Other biophysical elements, such as the salt concentration in the milieu and the presence of crowding agents, that exclude other molecules from a definite volume, can alter phase separation [60]. The inert crowding agent polyethylene glycol (PEG) is often used in vitro to enhance phase separation by mimicking a system with high viscosity. In this system, the effect of pH, temperature, and ionic strength can be investigated [61]. Next to synthetic crowding agents, biological agents such as BSA can be used. BSA has been shown to facilitate the condensation of single-stranded DNA-binding proteins (SSB) [62]. Temperature strongly impacts phase separation, and a change of only 1 °C has been shown to result in fused sarcoma (FUS) condensates forming or dissolving [2].

### 3.2. Chaperones

Molecular chaperons are essential for proteostasis and preventing aberrant folding and aggregation [63]. Chaperone-mediated regulation of phase separation has extensively been investigated in the context of stress assemblies, such as SGs. Chaperones are recruited to SG together with sequestered proteins and RNA that are not targeted for degradation and remain functional following chaperone-mediated dissolution [64,65]. Several studies have shown that chaperons influence the material properties of the condensate in which they partition [11,65–68]. Interestingly, many chaperons not only partition into biomolecular condensates [69–71], but also interfere with transient interactions necessary for other proteins to undergo phase separation, therefore affecting assembly/disassembly dynamics. This has been shown for the chaperone Hsp27 and the RNA-binding protein (RBP) FUS [70]. Analogously, phase separation of FUS is inhibited by transportin in neuron terminals [72].

### 3.3. RNA Helicases and RNA

DEAD-box proteins are RNA-dependent ATPases that unwind short duplexes of RNA [73]. In addition to their crucial role in regulating nearly all processes involving RNA, RNA-dependent DEAD-box ATPases (DDXs) have been shown to regulate phase separation and RNA flux into and out of biomolecular condensates. Elegant work demonstrated that ATP-bound DDXs promote condensation, whereas ATP hydrolysis induces condensates dissolution and release RNA [74–76]. The role of different types of RNA molecules in modulating condensate formation has been extensively investigated [77–81]. The concentration of RNA plays a crucial role in phase separation. When the RNA concentration is low, phase separation is favored. When the RNA concentration increases, the increasing negative charges will lead to a repulsion between molecules and induce condensate disassembly [82]. RNA can also modulate the material properties of condensates. For the condensates formed by the P-body-associated protein Dhh1 in presence of RNA in vitro, the RNA–protein interactions dominate over homotypic protein–protein interactions and decrease condensate dynamics [75]. It has been therefore suggested that RNA–RNA interactions might stabilize over time, thereby leading to a transition from a liquid-like to a solid-like state.

### 3.4. Posttranslational Modifications

An important tool to modulate intracellular phase separation is offered by post-translational modifications (PTMs). For proteins that partition into biomolecular condensates, PTMs can either promote mixing or demixing.

In the nucleus, SUMOylation of PML proteins promotes PML body nucleation by recruiting proteins containing SUMO interacting motifs (SIMs) [2,83,84]. In contrast, arginine methylation promotes disassembly of condensates formed by the RNA binding proteins (RBPs) hnRNPA2, Ddx4 and FUS [19,85,86]. This is due to the fact that arginine methylation weakens cation–π interactions that can be established between arginines and aromatic residues, such as tyrosines. Moreover, arginine methylation can disrupt protein–RNA interactions, and alter the nucleocytoplasmic transport of RBPs determining a change in RBPs concentration in nucleus and cytoplasm [87]. In different systems, arginine methylation has been shown to favor protein–protein interactions and condensate formation. This occurs for the protein Bucky ball (Buc) that contains a dimethylated tri-RG motif, and is essential for the formation of the Balbiani body, a condensate where organelles and RNAs are stored in oocytes [88]. Simmetrical dimethylation of SmD1 and SmD3 promotes assembly of snRNPs and spliceosomes [89]. Finally, dimethylation of FUS promotes assembly of Cajal body [90]. Arginines can be subject to citrullination, which is conversion of arginine to citrulline, catalyzed by the peptidyl arginine deiminase (PAD). The positively-charged guanidino group of an arginine residue will be converted to a neutral urea group, therefore changing the chemical properties of the protein. In line with this, citrullination of FUS was shown to abolish demixing in vitro [72].

Phosphorylation, a very well-studied form of PTM, can alter the number of transient interactions between condensates components [91,92]. In the case of Nephrin, tyrosine

phosphorylation promotes NCK recruitment and formation of condensates in vitro [16,22]. Similarly, phosphorylation induces formation of condensates involved in immune signaling [54], and nuclear proteins involved in gene silencing [44]. Other examples are the 5′-AMP-activated protein kinase-$\alpha$2 (AMPK-$\alpha$2) [93], and the mTOR effector kinases S6 kinase 1 and 2 (S6K1 and S6K2) that localize to and promote the formation of SGs [94]. Conversely, extensive work has shown that phosphorylation mediates the dissolution of multiple nuclear and cytoplasmic condensates [91]. The casein kinase 2 (CK2) mediates the phosphorylation of Ser-149 of G3BP1, a nucleator of SGs. As a consequence, G3BP1 oligomerization is impaired and SGs are disassembled [95]. Similarly, the spleen tyrosine kinase (Syk) is recruited to G3BP1-positive SGs and phosphorylates SG-localized proteins on tyrosine residues. This results in SGs' disassembly, although this mechanism is mediated by protein degradation via autophagy [96]. A prominent kinase that regulate SGs' assembly/disassembly, is the Dual-specificity tyrosine-regulated kinase 3 (DYRK3), a member of the human DYRK subfamily that plays a role in cell survival, differentiation, gene transcription, and endocytosis [97]. DYRK3 has been the first regulator of biomolecular condensates identified [7] and, since then, different studies have highlighted its role in promoting condensate dissolution by phosphorylation of multiple serine and threonine residues in unstructured domains [7,8,98]. In stressed cells, DYRK3 cycles between cytoplasm and SGs and directly phosphorylates the mTOR1 regulator PRAS40, thereby promoting SGs' dissolution, release of sequestered proteins and mTOR1 signaling activation [7]. More recently, the regulatory role of DYRK3 on biomolecular condensates has been investigated during the cell cycle [8]. During mitosis, in order to ensure optimal partitioning between the two daughter cells, both membrane-bound and membrane-devoid organelles that populate eukaryotic cells are required to become diffuse. It has been shown that DYRK3 interacts with multiple components of nuclear and cytoplasmic condensates, such as splicing speckles, pericentriolar satellites, and stress granules. Those DYRK3 interactors are found to be highly phosphorylated during mitosis. After nuclear envelope breakdown, DYRK3-to-interactor ratio increases to the point that condensate dissolution is induced. When DYRK3 activity is perturbed, aberrant hybrid condensates are formed and progression through cell cycle is blocked [8]. Moreover, DYRK3 has been found to modulate the dynamics of proteins that compose ERES (Endoplasmic Reticulum Exit sites), with consequences on anterograde vesicle trafficking [98]. Dual-specificity tyrosine-regulated kinases (DYRKs) are of particular interest, as they might have evolved to regulate a series of processes via their ability to modulate phase separation. In *C. elegans*, MBK-2, a member of the DYRK family, has been shown to promote the dissolution of P granules [99,100]. In mammalian cells, besides the DYRK3-mediated dissolution of condensates involved in stress recovery, cell cycle and membrane trafficking, other members of the DYRK family have been shown to regulate biomolecular condensates. This is the case of DYRK1A that regulates splicing speckles [101] and HIPK1 and HIPK2 that regulate PML nuclear bodies [102]. It is likely that the list of molecular regulators of phase separation will increase over the years.

*3.5. Membranes*

Although biomolecular condensates are not delimited by any membranes, in many cases, interactions between the two have been reported. Biomolecular condensates have been found in association with the plasma membrane [16,22,54,103–106], mitochondrial membranes [107,108], and the endoplasmic reticulum [98,109,110]. Membrane surfaces are emerging as regulatory elements that control biomolecular condensate formation [111], as they offer a two-dimensional system that reduce the critical concentration needed by molecules to undergo phase separation [22,103].

## 4. Pathological Functions of Biomolecular Condensates

As biomolecular condensates are implicated in multiple cellular processes, it is easy to imagine that alterations of the cellular capacity of modulating phase separation determines pathological states. This has been studied particularly in the case of debilitating

neurodegenerative diseases, where maturation of condensates to an irreversible solid state has been linked to the biogenesis of amyotrophic lateral sclerosis (ALS), frontotemporal dementia (FTD), and Alzheimer's disease (AD) [1,18,20,112–114]. Recently, the formation of aberrant condensates has been explored in other diseases, such as cancer and immune signaling [59,115]. These findings are summarized below.

### 4.1. Biomolecular Condensates in Immune Signaling

Dysregulation of phase separation has important consequences on anti-viral immune response. SARS-CoV-2 is a novel coronavirus that causes the disease known as the coronavirus disease 2019 (COVID-19) [116]. SARS-CoV-2 nucleocapsid (N) protein forms condensates that sequester RNA and the Ras-GTPase-activating protein SH3 domain-binding protein 1 (G3BP1). As a consequence, the interaction between G3BP1 and cGAS or RIG-I is suppressed, thereby inhibiting the anti-viral immune responses of the host cells [117]. Similarly, Herpes virus proteins ORF52 and VP22 compete with cGAS for DNA binding and disrupt cGAS condensates [118]. Epstein-Barr virus (EBV) is one of the most significant human tumor viruses and has been linked to various cancers [119,120]. EBV nuclear antigen 2 (EBNA2) and its coactivator EBNALP form condensates at active super enhancer sites of genes involved in cell growth and proliferation [121]. Many viruses form cytoplasmic inclusions in the host cells that are known as viral factories and are needed to escape host immune surveillance and promote genome replication and virus assembly. This has been reported for HIV1-nucleocapsid protein, and Vesicular stomatitis (VSV)- L, N, an P proteins [122,123]. For rabies virus RABV, the biomolecular condensates formed by viral proteins are represented by so-called Negri bodies and form via phase separation [124].

### 4.2. Biomolecular Condensates in Cancer

The speckle-type POZ protein (SPOP) is known to cause the formation of solid tumors, such as breast, prostate, colorectal, and gastric tumors [125,126]. SPOP acts as a substrate adaptor of the cullin3-RING ubiquitin ligase, thereby promoting substrate ubiquitination. However, cancer-associated mutations in SPOP interfere with substrate recruitment to the ligase and assembly of SPOP-substrates in nuclear speckles. Recently, it has been shown that SPOP undergoes phase separation, and this is driven by its dimerization and multivalent interactions with substrates, in vitro and in cells [127]. Disease-associated SPOP mutations interfere with its phase separation and co-localization in biomolecular condensates, and lead to the accumulation of proto-oncogenic proteins [127]. Similarly, mutations of the non-receptor protein tyrosine phosphatase SHP2 can lead to its accumulation in condensates. This, in turn, causes the hyperactivation of the RAS-mitogen-activated protein kinase (MAPK) signaling pathways, thereby promoting tumorigenesis [128]. Interestingly, phase separation of cancer-associated SHP2 mutants could be specifically attenuated by SHP2 allosteric inhibitors [128]. Moreover, phase separation of cancer-associated SRC-1, a previously known transcriptional coactivator for nuclear hormone receptors, could be selectively disrupted by the treatment of an anti-HIV drug elvitegravir (EVG), thereby suppressing oncogenic transcription [129]. In addition, AKAP95, a nuclear protein involved in splicing regulation, also forms condensates. Changes In the dynamics and fluidity of these condensates can significantly impact tumorigenesis [130].

Dysregulation of transcription is a hallmark of cancer [131]. Several transcriptional coactivators that are known to regulate tumorigenesis have been found to undergo phase separation. The transcriptional activators YAP and TAZ are downstream effectors of the Hippo pathway that, following a variety of signals, can activate the transcription of genes involved in various processes, including cell proliferation [132]. In cancer cells, YAP and TAZ form condensates that promote tumorigenesis and anti-PD1 immunotherapy resistance [133].

Recent findings on the oncogenic nucleoporin 98-homeobox A9 (NUP98-HOXA9) chimera, found in leukemia, revealed that this chimeric protein is able to drive aberrant

chromatin looping near leukemia-associated genes, further confirming the role of aberrant phase separation in tumor progression [134].

Conversely, loss of phase separation has been found in fibrolamellar carcinoma (FLC), a rare and lethal form of liver cancer [135].

Of particular interest is the discovery that anticancer drugs spontaneously partition into biomolecular condensates [59]. A recent study tested several compounds for their ability to concentrate into reconstituted nuclear condensates [136]. Among those are cisplatin and mitoxantrone, FDA-approved drugs that modify DNA through platination or intercalation [137] used in the treatment of different malignancies [138]. These compounds were able to concentrate in specific condensates even in the absence of their direct targets, thanks to π–π or cation–π interactions, suggesting that condensates form a physicochemical environment that favors drug concentration and influences pharmacodynamics. Tamoxifen, another FDA-approved antineoplastic compound used in the treatment of estrogen receptor (ER)-positive breast cancer [139], is also recruited to transcriptional condensates. However, tamoxifen recruitment to the condensates induces their disruption due to the displacement of ERα. This results in reduced oncogene expression in breast cancer cells [136]. Similarly, the over-the-counter medicines and small molecules lipoamide and lipoic acid [140,141] selectively dissolve SGs [142]. The effect of lipoamide/lipoic acid-mediated SG disruption has been investigated as a potential novel route to treat ALS, showing that it prevents dieback of ALS patient-derived motor neurons, and promotes recovery of motor functions in *Drosophila melanogaster*. Since SGs have been shown to regulate several oncogenic processes [143,144], interesting results might be obtained by testing the effect of lipoamide and lipoic acid in cancer cells. This evidence that a specific biomolecular condensate can be disrupted in a selective manner represents a novel and powerful area to investigate. Exciting results are provided by the identification of compounds that bind IDR of phase-separating molecules [145], and drugs that inhibit enzymes that impact condensate properties [8,98].

Taken together, these studies point towards a vital role of biomolecular condensates in tumor biology and highlight the opportunity of targeting phase separation to combat cancer.

### 4.3. Biomolecular Condensates in Neurodiseases

Under physiological conditions, biomolecular condensates are maintained in a liquid- or gel-like state that enables dynamic exchange of molecules. However, certain condensates have been found to mature over time and form irreversible aggregates. This is a hallmark of several neurodegenerative diseases, including frontotemporal dementia (FTD), Alzheimer's Disease (AD), amyotrophic lateral sclerosis (ALS), and Parkinson's disease (PD) [146,147]. Progression from reversible dynamic condensate to irreversible pathological aggregate has been shown for FUS, tau, TAR DNA-binding protein 43 (TDP43), and α-synuclein [20,148–150]. Condensate maturation to aggregate can be triggered and modulated by disease-associated PTMs and mutations. Examples are provided by altered methylation or phosphorylation on multiple positions of FUS [72,151], PD-associated S129 phosphorylation and E46K, and A53T mutations in α-synuclein [150], ALS-associated A321V mutation in TDP-43 [152]. A large body of work has focused on the crucial role played by biomolecular condensates in the onset and progression of neurodegenerative diseases, and this has extensively been reviewed elsewhere [114,153–155].

### 5. Concluding Remarks and Future Perspectives

A fundamental problem in cell biology is how the densely packed cellular space is organized in order to achieve spatiotemporal control over complex biochemical reactions. One way to achieve this is offered by classic organelles that are defined by surrounding lipid bilayer membranes. Conversely, biomolecular condensates lack physical barriers yet are able to maintain identity, concentrate molecules, and modulate biochemical activities. How this is possible remained elusive for many years. In the last decade, cellular and biochemical observations have been explained in light of polymer chemistry and soft matter

physics, providing a novel framework for understanding the biogenesis of biomolecular condensates [156]. Considerable work, both in vitro and in cells, has been done in order to identify the molecular components that can undergo phase separation and, among them, those that drive the formation of biomolecular condensate [2,3]. Numerous studies have focused on uncovering the sequence-encoded molecular grammar and the type of specific interactions promoting phase separation of specific biomolecules [21,157]. Moreover, a large number of studies have demonstrated the significance of biomolecular condensates in orchestrating crucial cellular processes. Dysregulation of these cellular processes is a key event in initiation and/or evolution of diseases. For instance, aberrant phase separation is exploited by viruses to escape cellular antiviral response [115]. Similarly, aberrant phase separation has been described in a large number of cancers [158]. Importantly, intracellular phase separation not only provides a useful new framework to understand complex immune responses and/or oncogenic pathways, but also presents an exciting novel approach to target intractable and undruggable proteins. A future challenge for the field is the identification of more specific drugs that inhibit the formation of aberrant condensates as potential new therapeutic treatments [59].

Regulation of biomolecular condensates can be mediated by a large number of factors, such as biophysical parameters, endomembranes, chaperons, RNA helicases, and PTMs. Among PTMs, serine and tyrosine phosphorylation by kinases and dephosphorylation by phosphatases offer a reversible and rapid switch to control biomolecular condensates integrity and dynamics, via modulation of protein valency and interaction strength [91,92]. Interestingly certain condensates are able to exclude or enrich specific enzymes. An example is provided by T-cell condensates that generate a compartment with physicochemical properties able to exclude phosphatases and recruit kinases [54]. In this system, kinase-mediated phosphorylation promotes condensates growth, signaling activation, and recruitment of actin regulators that facilitate actin filament assembly. In contrast, for different biomolecular condensates, kinase recruitment and phosphorylation of targets result in disassembly of the compartment [91]. Although many molecular regulators of biomolecular condensates have been identified to date, it is likely that their number will expand in the future years. Moreover, many questions remain concerning how these regulators are recruited to their target condensate and how they are activated/inactivated. Interesting insights are provided by DYRK3, the first identified molecular regulator of phase separation in cells, and well-studied kinase in the context of biomolecular condensates [7,8,98]. In particular, DYRK3 can phase separate at a specific concentration threshold, and has been shown to partition into the target condensate in a mechanism that seems to be mediated by its N-teminus. However, after reaching the target condensate DYRK3 can cycle between this structure and the surrounding nucleo- or cyto-plasm. Newly synthesized DYRK kinases have been shown to autophosphorylate in order to reach the correct folding [159], and this ability might be retained by the fully mature kinase [160]. Therefore, we can speculate that DYRK3 cycling behavior might be promoted by an autophosphorylation event, although this remains to be experimentally proved. In the cytoplasm, DYRK3 is stabilized by HSp90, and therefore protected from degradation [71]. Interestingly, DYRK3 has been shown to exert control over different kinds of cytoplasmic and nuclear biomolecular condensates in various physiological conditions. It would be interesting to investigate whether DYRK3 plays a role in pathological states and can be exploited to dissolve aberrant condensates found in many diseases. More broadly, phosphorylation and dephosphorylation events could regulate not only condensate assembly/disassembly, but also alter the strength of the interactions that mediate partitioning of regulating enzymes into their target condensate.

It is becoming evident that not only presence/absence of biomolecular condensate, but also their material properties, impact condensate function [18,74,75,98,155,161]. It is believed that, over time, liquid-like condensates mature to more stable states that are sometimes beneficial in cases where a certain level of rigidity is required to form a stable structural matrix [17,162–164]. However, this liquid-to-solid phase transition can also result into formation of fibrous structures that are detrimental to the cell functioning, as

it has been particularly investigated in neurodegenerative diseases [114,153,154]. Many different factors can alter the material properties of condensates, such as mutations in condensate components or altered regulation of the interactions driving demixing of molecules [21,74,75,92,98]. Shedding light on how living cells regulate the material properties of biomolecular condensates and how this regulation is altered in pathological states remain of crucial relevance.

**Funding:** R.G. is supported by POR Calabria FESR/FSE 20142020 (D.D.R.C. n. 4584 del 04.05.2021).

**Institutional Review Board Statement:** Not applicable.

**Informed Consent Statement:** Not applicable.

**Data Availability Statement:** Not applicable.

**Conflicts of Interest:** The authors declare no conflict of interest.

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
