# Peer review of "Intracellular Organization of Proteins and Nucleic Acids via Biomolecular Condensates in Human Health and Diseases"

_2673-6411, doi:10.3390/biochem3010003_

Round 1
Reviewer 1 Report
The manuscript entitled “Intracellular organization of proteins and nucleic acids via biomolecular condensates in human health and diseases” by Gallo et al nicely describe the current research on various biomolecular condensates, their physiological and pathological functions and future perspectives. I find this brief description is interesting to a wide range of reader having an overview of biomolecular condensates and phase separation. I am recommending this work for publication after some correction/changes.
Comments:
1. Can the author include some more recent references or information on current FDA approved/ in clinical trial drugs discovered based on biomolecular condensates?
2. Schematic represents of biomolecular condensates involved in different patholigical diseases will be helpful for general reader of this journal for better overview of condensates.
3. Can the author describe the role of biomolecular condensates in neurodiseases?
4. Also, include a schematic diagram with a brief description of different interaction involved in phase separation and how those interactions favors/regulates formation of different biomolecular condensates.
Author Response
The manuscript entitled “Intracellular organization of proteins and nucleic acids via biomolecular condensates in human health and diseases” by Gallo et al nicely describe the current research on various biomolecular condensates, their physiological and pathological functions and future perspectives. I find this brief description is interesting to a wide range of reader having an overview of biomolecular condensates and phase separation. I am recommending this work for publication after some correction/changes.
We thank the reviewer for taking the time to review this manuscript. Moreover, we thank the reviewer for appreciating this manuscript and for all the helpful suggestions. We have addressed their comments in the revised manuscript as detailed below. Overall, we believe that the changes that have been made in the current version of this manuscript have greatly improved its quality.
Comments:
- Can the author include some more recent references or information on current FDA approved/ in clinical trial drugs discovered based on biomolecular condensates?
In section 4.2 (lines ) information about FDA approved drugs found in condensates and the relevant references have been provided.
- Schematic represents of biomolecular condensates involved in different patholigical diseases will be helpful for general reader of this journal for better overview of condensates.
We have summarized the biomolecular condensates found in health and pathological states in Figure 2.
- Can the author describe the role of biomolecular condensates in neurodiseases?
A short new section has been included (section 4.3) and we report the major findings regarding the role of biomolecular condensates in neurodegenerative diseases. Moreover, relevant recent references are reported.
- Also, include a schematic diagram with a brief description of different interaction involved in phase separation and how those interactions favors/regulates formation of different biomolecular condensates.
We have added Figure 1 that reports schematic diagram of interactions involved in phase separation.

Reviewer 2 Report
Author Comments
The manuscript entitled "Intracellular organization of proteins and nucleic acids via biomolecular
condensates in human health and diseases" elucidated the trendy understanding of intracellular phase
separation and described its biological functions and how this process has been regulated in eukaryotic
cells. The author has significantly contributed to explaining the importance of recent advances in
biomolecular condensates in developing pathophysiological conditions, particularly Cancer and immune
signaling. Overall, this is a clear, concise, and well-written review. The introduction is relevant and
theory-based. Sufficient information about the previous study findings is presented for readers to follow
the present study rationale. The inclusion of literature is generally appropriate, although adding a few
details specifically in the Post-translational section should be provided. Though the review is written well,
some modifications in a few sentences and adding some details and suitable references could improve it.
My comments here are concerned solely with the organization of the manuscript. Considering these points
will lead to an improved report that illustrates the key concepts and conclusions better.
1-Author: Line 23-25- A powerful tool for the intracellular.... well-studied phenomenon in polymer
chemistry called phase separation.
Comment: The author must cite some relevant references here.
2-Author: Line 44-49-Of those macromolecules.....the multivalent scaffolds
Comment: The author has introduced the terms "scaffold and client," which need to promote condensate
formation. Could you please provide some specific examples of both categories involved? This would
improve the statement further.
2. As this is a crucial part of the current topic, please provide a more detailed and precise difference
between those two categories (scaffold and clients) in terms of their interaction and their recruitment for
the formation of condensates.
3. Author: Lines 81-82-
Comment: Give an example of those proteins.
4. Author: Lines 85-87
Comment: Please provide a relevant reference for this statement.
5. Author: Lines 95-98-Transcription condensates in the nucleus are often found at the promoter region
and at super-enhancers, comprise of transcription factors and coactivators [40]–[43], and can regulate the
transcription of key genes that determine cell identity [36], [44].
Comment: Please provide an example of such genes here, as this is an integral part of transcription
regulation. Adding a few details could improve the statements in this section.
6. Author: Lines 114-116- The resulting condensates are... anti-viral immune responses.
Comment: Please provide a relevant reference for this statement.
2. Could you please explain the type of viral infection or provide some examples of such viruses against
whom these condensates show antagonistic responses?
7. Author: Lines 125-129- Other biophysical elements...condensated forming or dissolving [51].
Comment: The author has described some biophysical elements, such as salt and temperature. Do other
factors, such as volume, pH, and co-solutes, affect the phase separation process? If yes, please explain
here and add to the current section.
2. The author has explained the presence of the crowding agents. Could you please mention the examples
of some crucial crowding agents frequently used in phase separation and add them here?
3. Besides the synthetic polymers, are there any biological crowders (proteins) used as a common crowing
agent? If Yes, could you please add a few sentences regarding the same? This could improve the
familiarity of using synthetic and biological crowding agents in this section.
8 . Author: Section 3.4. Post-translational Modification
Comment: Though the author has explained all the crucial PTMs here, Arg-mediated methylation is one
of the essential PTMs affecting intracellular phase separation. Could you give more detailed information
about its less dynamic nature than other PTMs? Why it acts as a "rapid switch" in response to cellular
environmental change.
9 . Author: Section 4.2. Biomolecular condensates in Cancer, Lines 216-221
Comment: Please provide the relevant reference here.
Comment- Though the authors have described all sections very well, the conclusion paragraph should be better discussed and consist of some modifications regarding the future advantages in developing pathophysiological conditions, specifically for cancer and anti-viral immune signaling.
Author Response
Author Comments
The manuscript entitled "Intracellular organization of proteins and nucleic acids via biomolecular
condensates in human health and diseases" elucidated the trendy understanding of intracellular phase separation and described its biological functions and how this process has been regulated in eukaryotic cells. The author has significantly contributed to explaining the importance of recent advances in biomolecular condensates in developing pathophysiological conditions, particularly Cancer and immune signaling. Overall, this is a clear, concise, and well-written review. The introduction is relevant and theory-based. Sufficient information about the previous study findings is presented for readers to follow the present study rationale.
We thank the reviewer for taking the time to review the original manuscript. Moreover, we thank the reviewer for appreciating important parts of this manuscript, as well as for pointing out critical aspects, and offering very helpful suggestions. We have addressed their comments in the revised manuscript as detailed below. Overall, we believe that the changes that have been made in the current version of this manuscript have greatly improved its quality and readability.
The inclusion of literature is generally appropriate, although adding a few details specifically in the Post-translational section should be provided. Though the review is written well, some modifications in a few sentences and adding some details and suitable references could improve it.
My comments here are concerned solely with the organization of the manuscript. Considering these points will lead to an improved report that illustrates the key concepts and conclusions better.
1-Author: Line 23-25- A powerful tool for the intracellular.... well-studied phenomenon in polymer
chemistry called phase separation.
Comment: The author must cite some relevant references here.
Appropriate references have been added.
2-Author: Line 44-49-Of those macromolecules.....the multivalent scaffolds
Comment: The author has introduced the terms "scaffold and client," which need to promote condensate formation. Could you please provide some specific examples of both categories involved? This would improve the statement further.
2. As this is a crucial part of the current topic, please provide a more detailed and precise difference
between those two categories (scaffold and clients) in terms of their interaction and their recruitment for the formation of condensates.
Examples of both categories have been added to the text and we believe it improved the clarification of the distinction between scaffolds and clients.
Author: Lines 81-82-
Comment:Give an example of those proteins.
Examples of proteins that use phase separation to buffer their concentrations have been provided.
Author: Lines 85-87
Comment: Please provide a relevant reference for this statement.
Relevant references have been provided.
Author: Lines 95-98-Transcription condensates in the nucleus are often found at the promoter region and at super-enhancers, comprise of transcription factors and coactivators [40]–[43], and can regulate the transcription of key genes that determine cell identity [36], [44].
Comment:Please provide an example of such genes here, as this is an integral part of transcription
regulation. Adding a few details could improve the statements in this section.
Relevant examples have been added in this section.
Author: Lines 114-116- The resulting condensates are... anti-viral immune responses.
Comment:Please provide a relevant reference for this statement.
2. Could you please explain the type of viral infection or provide some examples of such viruses against whom these condensates show antagonistic responses?
We added a few sentences (and references) to clarify the anti-viral immune responses activated. We provided an example of viruses were condensates can play a protective effect. Moreover, we extended the examples of condensates that seem to be exploited for viral infection progression. However the latter information has been included in section 4.1 for continuity.
Author: Lines 125-129- Other biophysical elements...condensated forming or dissolving [51].
Comment:The author has described some biophysical elements, such as salt and temperature. Do other factors, such as volume, pH, and co-solutes, affect the phase separation process? If yes, please explain here and add to the current section.
2. The author has explained the presence of the crowding agents. Could you please mention the examples of some crucial crowding agents frequently used in phase separation and add them here?
3. Besides the synthetic polymers, are there any biological crowders (proteins) used as a common crowing agent? If Yes, could you please add a few sentences regarding the same? This could improve the familiarity of using synthetic and biological crowding agents in this section.
We have briefly included the requested examples of synthetic and biological crowding agents in this section.
8 . Author: Section 3.4. Post-translational Modification
Comment: Though the author has explained all the crucial PTMs here, Arg-mediated methylation is one of the essential PTMs affecting intracellular phase separation. Could you give more detailed information about its less dynamic nature than other PTMs? Why it acts as a "rapid switch" in response to cellular environmental change.
Section “3.4 Posttranslational modifications” has been extended and more details have been provided. In particular, the effect of phosphorylation on different condensates has been extensively discussed and the appropriate literature had been cited. The paragraph reporting findings on arginine-methylation has been extended, we provided examples of condensates where this PTM has been found and its effects.
9 . Author: Section 4.2. Biomolecular condensates in Cancer, Lines 216-221
Comment: Please provide the relevant reference here.
Appropriate reference has been provided.
Comment- Though the authors have described all sections very well, the conclusion paragraph should be better discussed and consist of some modifications regarding the future advantages in developing pathophysiological conditions, specifically for cancer and anti-viral immune signaling.
We thank the reviewer for point out the weaknesses of the discussion. We have extended this section.

Round 2
Reviewer 1 Report
Dear Authors,
Corrections/modifications of this manuscript are looks okay to me and I recommend this article to be accepted in its current form.
Thanks
Reviewer 2 Report
The authors have addressed all the comments and questions. I believe the manuscript is suitable for publication.